# Assessment of the Diffuse Attenuation Coefficient of Photosynthetically Active Radiation in a Chilean Lake

**Lien Rodríguez-López** [1]**, Lisdelys González-Rodríguez** [2,]*****, Iongel Duran-Llacer** [3]**, Wirmer García** [4]**, Rolando Cardenas** [5] **and Roberto Urrutia** [6]

[1] Facultad de Ingeniería, Arquitectura y Diseño, Universidad San Sebastián, Lientur 1457, Concepción 4030000, Chile
[2] Facultad de Ingeniería y Negocios, Universidad de Las Américas, Sede Concepción, Región del Biobío, Concepción 4030000, Chile
[3] Hémera Centro de Observación de la Tierra, Facultad de Ciencias, Ingeniería y Tecnología, Universidad Mayor, Camino La Pirámide 5750, Huechuraba, Santiago 8580745, Chile
[4] Programa de Magíster en Ecología Marina, Departamento de Ecología, Facultad de Ciencias, Universidad Católica de la Santísima Concepción, Concepción 4030000, Chile
[5] Facultad de Matemática- Física- Computación, Universidad Martha Abreu de las Villas, Santa Clara 54830, Cuba
[6] Facultad de Ciencias Ambientales, Universidad de Concepción, Concepción 4070386, Chile
***** Correspondence: lgonzalezr@udla.cl; Tel.: +56-999-471-803

**Abstract:** The diffuse attenuation coefficient of photosynthetically active radiation is an important inherent optical property of the subaquatic light field. This parameter, as a measure of the transparency of the medium, is a good indicator of water quality. Degradation of the optical properties of water due to anthropogenic disturbances is a common phenomenon in freshwater ecosystems. In this study, we used four algorithm-based Landsat 8 OLI and Sentinel-2A/B MSI images to estimate the diffuse attenuation coefficient of photosynthetically active radiation in Lake Villarrica located in south-central Chile. The algorithms' estimated data from the ACOLITE module were validated with in situ measurements from six sampling stations. Seasonal and intralake variations of the light attenuation coefficient were studied. The relationship between the diffuse attenuation coefficient of photosynthetically active radiation, meteorological parameters, and an optical classification was also explored. The best results were obtained with QAA v6 KdPAR Nechad ($R^2 = 0.931$, $MBE = 0.023$ m$^{-1}$, $RMSE = 0.088$ m$^{-1}$, and $MAPE = 35.9\%$) for spring and QAA v5 Kd490 algorithms ($R^2 = 0.919$, $MBE = -0.064$ m$^{-1}$, $RMSE = -0.09$ m$^{-1}$, and $MAPE = 30.3\%$) for summer. High KdPAR values are associated with the strong wind and precipitation events suggest they are caused by sediment resuspension. Finally, an optical classification of freshwater ecosystems was proposed for this lake. The promising results of this study suggest that the combination of in situ data and observation satellites can be useful for assessing the bio-optical state of water and water quality dynamics in Chilean aquatic systems.

**Keywords:** Secchi disk optical depth; diffuse attenuation coefficient; landsat and sentinel images; Lake Villarrica; Chile





## 1. Introduction

Hydrologic optics is concerned with the behavior of light in aquatic media [1]. Underwater light attenuation plays an important role in modulating aquatic ecosystems and is considered a sentinel of climate change and human activity [2,3]. Changes in underwater light attenuation can indicate shifts in water quality, including acidification [4], increased terrestrial loading of organic carbon [5], or algal blooms [6]. A parameter that helps in ascertaining phytoplankton and sediment concentrations is the diffuse attenuation coefficient of photosynthetically active radiation (KdPAR, m$^{-1}$), which estimates water transparency or turbidity by measuring the penetration capacity of the solar

photosynthetically active radiation (PAR; 400–700 nm) incident on it. Solar radiation is crucial to nearly all of Earth's ecosystems, including dynamics of freshwater ecosystems. Practically all the energy that drives and controls lake metabolism derives directly from solar energy [7]. With KdPAR, it is possible to estimate the depth of the euphotic layer, which corresponds to the depth in the water column where PAR decreases to 1% intensity relative to its surface magnitude [8,9]. In the literature, KdPAR was estimated directly using irradiance sensors measuring the flow of PAR photons in the water column [10,11], or indirectly, using the Secchi disk (SD) [12]. Due to the high costs of quanta meters, KdPAR is usually predicted from the SD in situ measurement using regression analyses of these parameters, which provides empirical constants. According to [1], the empirical constant represents the optical depth (OD) at which an SD disappears underwater. Studies report empirical values between 1.70 for clear water and 1.44 for turbid water [13], while [14] proposed estimating KdPAR with an OD of 2.

KdPAR, when correlated with the SD depth, provides the means to physically categorize water according to color. Its color can be interpreted as a measure of water turbidity or transparency and has been a valuable tool in several studies [15–18]. Measurements of KdPAR are important to identify the optimal zones for the development of photosynthetic activity and therefore aquatic life.

Oceanographer N. Jerlov developed an optical classification of ocean and coastal water, using the attenuation coefficients for light, mostly in the range of 310–700 nm [19]. He proposed three types of ocean water (I, II, and III), which later were extended to include intermediate types IA and IB. For coastal water, five types were proposed (C1, C3, C5, C7, C9). In both cases, turbidity increases with the type number. However, the works of Jerlov were not extended to freshwater ecosystems. Therefore, an optical classification of freshwater ecosystems would be very novel and useful.

A factor of major importance in the hydrodynamics of lakes is wind, which generates currents and mixing in the water mass. Such movements not only influence the distribution and aggregation of nutrients, but also the distribution of microorganisms and plankton. Rodríguez-López [18] indicated that meteorological conditions, mainly precipitation and wind speed, may affect the variation of turbidity and water clarity of lakes. Similarly, in Ref. [20] many covarying points were found between KdPAR and maximum wind speed in observations made during the years 2003–2014. Hence, high KdPAR values can be associated with the strong wind and precipitation event suggested by sediment resuspension.

Information from satellite remote sensing is increasingly being used to complement data from in situ monitoring networks. Various recent studies have used remote sensing as a tool to monitor water quality parameters such as transparency, chlorophyll-a, and turbidity [21–24], including KdPAR [25,26]. Using remote sensing, it is possible to estimate and track the spatial and temporal variability of KdPAR. More recently, several algorithms have been developed to find KdPAR based on spectral remote-sensing reflectance [27,28]. The standard quasi-analytical algorithm (QAA) [29] was first tuned with Visible Infrared Imaging Radiometer Suite ocean color sensor bands such as QAA-V and then extended to several satellite sensors (Landsat, Sentinel, and others) to obtain more accurate estimates of the inherent optical properties (IOP) of water (e.g., backscattering measurements) in shallow estuarine and nearshore water. A recent version (QAA_v6) was presented online by Lee [30], while other algorithms developed for Nechad versions 5 and 6 have been less studied and are included in the present study. Others such as [31] have adapted the algorithms for hydroelectric reservoir applications. The QAA has been extensively validated using simulated and field data sets from different geographic regions [28,32–35]. Therefore, global and regional coverage of KdPAR at high spatial and temporal resolution can be provided by algorithms based on satellite data, and this coverage can be used to improve our understanding of the physical, chemical, and biological processes in lakes.

Chile has several lake districts that play an important role as freshwater reservoirs and in the provision of multiple ecosystem services. Some studies have suggested that the

Landsat and Sentinel satellites provide good results for estimated quality water parameters in Chilean lakes [24,36]; nonetheless, lake monitoring in Chile is insufficient due to resource and research limitations. According to [37], barely 5% of lakes are monitored by the General Water Directorate of Chile (DGA for its abbreviation in Spanish). Lake Villarrica is an Araucanian Lake with marked human activity in its surroundings [38,39]; therefore, it is part of an Environmental Quality Program for the Protection of Inland Surface Waters. Algal blooms frequently occur in some regions of this lake [6]. A characteristic phenomenon of the lake area is the so-called Puelche wind [40], which causes intense swells in the lake shore area. Although Lake Villarrica is one of the most studied Chilean lakes and a secondary water quality standard for it has been established [41], studies on KdPAR characteristics lack coverage. Thus, there is still much work to be done in this area.

The hypothesis to be tested in this research is whether SD-derived KdPAR could be related to a specific optical water type (Jerlov's classification), and, in turn, could be derived from satellite data that would broaden the application of the SD approach and increase its usefulness in understanding the spatio-temporal distribution of KdPAR in Lake Villarrica. Thus, the objectives were to (i) apply and validate algorithms that retrieve the KdPAR, (ii) analyze the seasonally spatio-temporal distribution of KdPAR in Lake Villarrica, and (iii) optically classify the inland aquatic ecosystem following Jerlov's classification.

## 2. Materials and Methods

### 2.1. Study Area

Figure 1a shows Chile in South America, while Figure 1b shows the location of Lake Villarrica in the 9th, or Araucanía Region, between 39°18′S and 72°05′W at 230 m.a.s.l (Figure 1c). The lake is among the largest bodies of water in Chile. Its hydrographic basin has a surface area of 2920 km$^2$ and the water mirror is 71 km$^2$. The maximum depth of the lake is 167 m. Its maximum length is 23 km and its maximum width is 11 km. The water renewal period is 2 to 4 years [42]. Its main tributary is the Pucón River, and its effluent is the Toltén River [42]. On the eastern and southern shores of the lake, there are contributions of material from the Trancura River and Villarrica Volcano. This river collects the water from mountain snowmelt and Caburgua Lake. The area has the characteristics of a rainy temperate oceanic climate. The lake has a regular coastline with few sheltered bays. Its depth increases significantly just a few meters from the shore. The most notable bays are Villarrica Bay and the wide Pucón Bay. The surface temperature reaches a range of 19 °C to 22 °C in summer, while in the winter it fluctuates between 9 °C and 10 °C. Precipitation occurs throughout the year and the thermal amplitude increases towards the interior. Frosts are frequent in the winter season. The average annual rainfall is 1465 mm. The cities of Villarrica and Pucón are very touristic areas, resulting in increases in capital flows, investments, rentals, real estate development, and migration [43,44]. Recent studies on the lake found high nutrient content—a result of the multiple uses of its basin—which could lead the lake to a trophic state transition; therefore, a secondary water quality standard was established to protect the lake [41].

### 2.2. Monitoring Campaign and Field Data Collection

The in-situ measurements in the 2000–2021 period were carried out in three monitoring campaigns. The first was reported in "Redefinition of the minimum network of lakes" by POCH AMBIENTAL S.A. in the study "Final report No. 1. Phytoplankton analysis in water samples. Tender No. 1019-98-LE13." The second was reported in the study "Analysis and reformulation network monitoring lakes region of the river" (DGA, 2020), and the third was carried out by the Eula Center in 2021. Secchi disk (SD) depth, chlorophyll a (Chl-a), and turbidity (NTU) field data were collected at 6 sampling stations (V1–V6) (see Figure 1c).

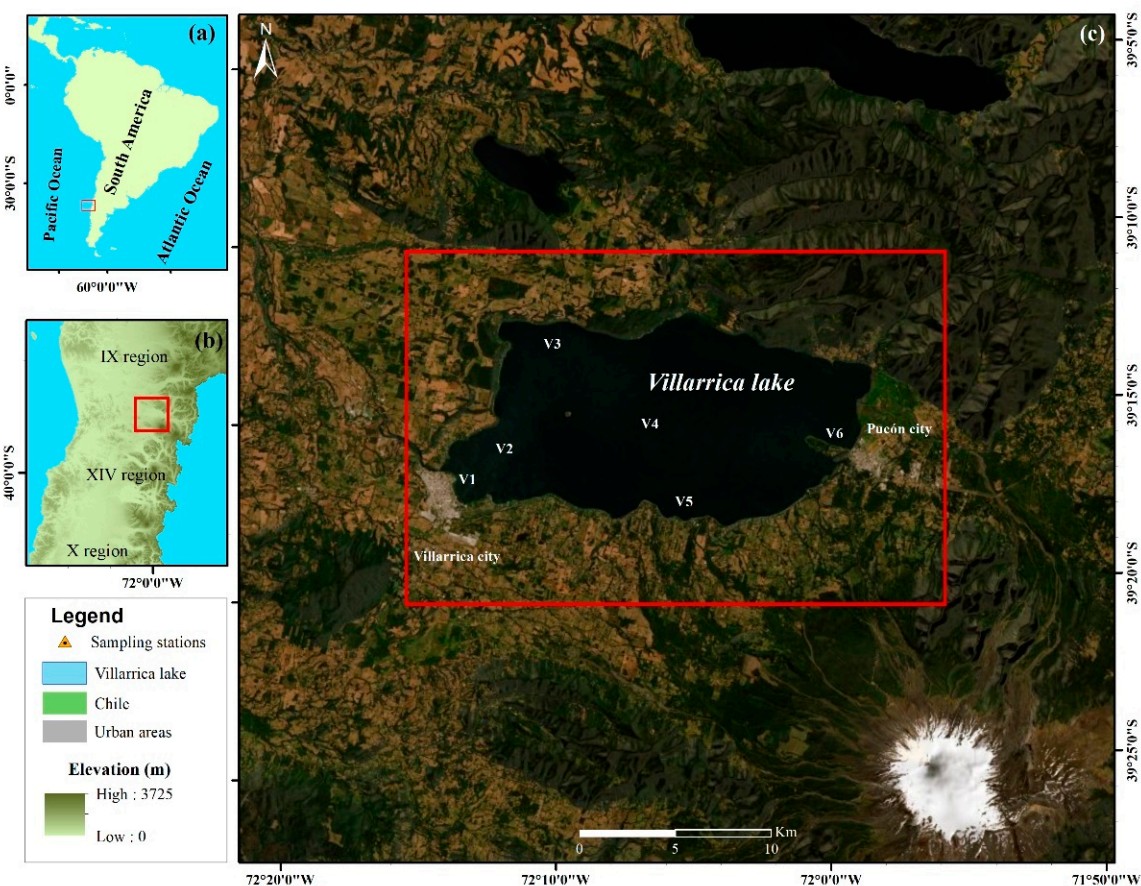

**Figure 1.** (**a**) Chile in South America, (**b**) location of Lake Villarrica in the 9th Region, and (**c**) satellite image of Lake Villarrica with sampling stations V1–V6, represented by yellow triangles. The grey areas represent the cities on the lake shore. Satellite image extracted from Landsat 8.

A monitoring station with radiometric and meteorological sensors was installed. Table S1 shows the main characteristics of the radiometric sensor. In addition, it included a Campbell Scientific CR300 data logger and temperature (model 107-L20) and relative humidity sensors (model HMP60-L11). Global solar radiation (IG, Wm$^{-2}$), cloud frequency (%), and air temperature (T, $^\circ$C) variables were obtained from the Solar Explorer [45]. Datasets for the 2004–2016 period were downloaded from http://solar.minenergia.cl/inicio, accessed on 15 January 2022, while wind speed (WS, ms$^{-1}$) for the 1980–2017 period was obtained from https://eolico.minenergia.cl/exploracion, accessed on 1 February 2022. In addition, hourly WS, wind gust, and wind direction records for 2017–2020 were downloaded from the Instituto de Investigaciones Agrarias and Dirección Meteorológica de Chile at https://climatologia.meteochile.gob.cl/, accessed on 15 January 2022. Precipitation (rain, mm) for the 1984–2020 period was obtained from the Center for Climate and Resilience Research (available at: http://explorador.cr2.cl, accessed on 1 February 2022) [46]. The wind forcing and precipitation values were taken in a three-day window (72 h) before the image acquisitions.

### 2.3. Diffuse Attenuation Coefficient and Optical Water Classification

To obtain KdPAR empirical formulas Equations (1)–(3), in accordance with [18,47] were used. Thus, average KdPAR was calculated for each sampling station (V1–V6) and season (summer and spring). In addition, to propose an optical classification of this lake ecosystem, we compared the KdPAR according to Jerlov's classification [19].

$$z(\text{ph}) = \frac{4.6}{\text{KdPAR}} \tag{1}$$

$$z(\text{ph}) = 2.3 \times \text{SD} \tag{2}$$

Combining the above two formulae:

$$\text{KdPAR} = \frac{2}{\text{SD}} \tag{3}$$

where: SD is Secchi disk depth, and z(ph) is the photic zone.

### 2.4. Landsat and Sentinel Images

A total of 14 Landsat 8 Operational Land Imager (L8/OLI) multispectral images with a 30 × 30-m resolution was obtained from the Department of the Interior U.S. Geological Survey (USGS). The L8 mission, with a revisit frequency of 16 days, acquires images of Earth's terrestrial and polar regions in the visible, near-infrared (NIR), shortwave, and thermal infrared spectra [48]. The OLI onboard L8 has enhanced spectral characteristics with an improved signal-to-noise ratio and 12-bit radiometric resolution [49,50]. Orthorectified and terrain-corrected Collection 2 Level 1 images were downloaded free of charge from the official site of the United States Geological Survey (USGS) Earth Explorer (https://earthexplorer.usgs.gov/, accessed on 8 December 2021). The tiles corresponding to the region of interest were in paths 232/233 and row 87 (see Table S2 for acquisition dates).

In addition, 22 Sentinel 2 (S2 A/B) images were selected for this study. The Sentinel fleet is a collaboration between the European Commission and the European Space Agency (ESA, https://earth.esa.int/, accessed on 8 December 2021), developed to complement the operational needs of the Copernicus program. The S2 A/B Multispectral Instrument (MSI) mission, with a revisit frequency of 5 days, is based on a constellation of two identical satellites in the same orbit, 180° out of phase with each other. The radiometric resolution of the MSI onboard S2 is 12 bits. The S2 scenes for the study area were downloaded from the Sentinel Science Data Center (https://scihub.copernicus.eu/dhus/#/home, accessed on 8 December 2021) thanks to its open and free data access policy. These images are radiometrically and geometrically corrected Level 1C (L1C) products in the top of atmosphere (TOA). For our study area, the images used belong to the T18HYB and T19HBS tiles (HYB and HBS subtitles). For both L8 and S2, only scenes with low cloud cover (less than 12%) over the study region were selected for further analysis.

All the L8 and S2 (L1C) images were atmospherically corrected to bottom of atmosphere (BOA) with the ACOLITE program (version 20211124.0) (https://github.com/acolite, accessed on 1 January 2022). ACOLITE is a generic processor that was specifically developed for marine, coastal, and inland water and supports free processing of both L8 and S2 [51]. The default atmospheric correction using the "Dark Spectrum Fitting" (DSF algorithm) approach was selected and used in the ACOLITE processor [52–54]. The ACOLITE products obtained correspond to surface-level reflectance (Rrs) for L8 and S2.

### 2.5. Algorithms for KdPAR Estimation

ACOLITE includes several algorithms for the retrieval of parameters derived from reflectance such as KdPAR. The products and algorithms used for both satellite missions (L8 and S2) were as follows: (1) QAA v5 Kd490 nm [27]; (2) QAA v6 KdPAR Lee [30]; (3) QAA v5 KdPAR Nechad; and (4) QAA v6 KdPAR Nechad. A QAA derives the absorption and backscattering coefficients of water from satellite images [29] and then determines KdPAR/Kd490 based on its relationships with the absorption and backscattering coefficients. The v5 or v6 specification denotes version 5 or 6 outputs. As the result of the image processing, the mean pixels in a 3 × 3 window were extracted with SNAP version 8.0 (https://step.esa.int/main/download/snap-download/, accessed on 1 March 2022). All algorithms were used to represent and understand the spatio-temporal distribution of KdPAR in summer and spring.

*2.6. Statistical Analysis*

We used an in situ dataset and the preprocessed Landsat 8 and Sentinel images that cover the 2017–2021 period. The in situ dataset includes four physic-chemical parameters related to the transparency of the lake (Chl-a, NTU, SD, and SD-derived KdPAR), measured at six sampling stations. Statistical analyses, including calculations of the average, maximum, and minimum values, were performed using Origin Pro Academic software. In addition, the coefficient of variation (CV), standard deviation (SD), and number of data (*n*) were calculated. The accuracy of algorithms was assessed by calculating the coefficient of determination ($R^2$), while the mean absolute percentage error (*MAPE*), the root-mean-square error (*RMSE*), and the mean bias error (*MBE*) were used to assess the accuracy of the model performances for validation processes. In addition, we applied a Taylor diagram [55], which can represent three different statistics simultaneously (i.e., the centered *RMSE*, the Pearson correlation, and the standard deviation). Taylor diagrams provide a visual framework for comparing a suite of variables from one or more test data sets to one or more reference data sets [56]. The details on how to calculate these statistical indicators can be found in [18,24].

## 3. Results

*3.1. Monitoring Campaign and Field Data Collection*

Observations carried out in Lake Villarrica from 2017 to 2021 indicate that the monthly average daily PAR values were 677.04 mmol/m$^2$, ranging from a summer maximum of 1237.38 mmol/m$^2$ (December) to a winter minimum of 181.04 mmol/m$^2$ (June) (Table 1). Similarly, the monthly average daily air temperature value was 11.84 °C, also varying from a summer maximum of 16.64 °C (February) to a winter minimum of 7.39 °C (July). In general, cloudiness increases in the winter months, resulting in PAR decreasing, and vice versa during the summer months. The mean cloud cover in the area was 23.67%, with a winter maximum of 26.76% (August) and a summer minimum of 17.37% (February). Based on hydrometric statistics, the total rainfall during the entire study period was 6210.0 mm, ranging from a maximum value of 13,154.4 mm (June) to a minimum value of 2164.8 mm (February). There is no dry season in this region, but there is a significant decrease in precipitation during summer. The surface wind pattern during winter was mainly northwest and north, with a maximum speed of 5.82 ms$^{-1}$, and during summer it was mainly southwest and west, with a minimum speed of 2.68 ms$^{-1}$. The relative humidity was high across the whole study period (>66.58%).

**Table 1.** Meteorological and radiometric parameters in Lake Villarrica.

| Month. | T (°C) | WS (ms$^{-1}$) | RH (%) | Rain (mm) | PAR (mmol/m$^2$) | Cloud Cover (%) |
|--------|--------|--------|--------|-----------|------------------|-----------------|
| Jan | 16.62 | 2.68 | 69.82 | 2521.7 | 1235.37 | 18.69 |
| Feb | 16.64 | 2.78 | 66.58 | 2164.8 | 1082.19 | 17.37 |
| Mar | 14.90 | 2.79 | 73.79 | 3397.0 | 764.56 | 20.47 |
| Apr | 11.83 | 3.33 | 84.08 | 5586.7 | 436.46 | 21.89 |
| May | 9.74 | 3.76 | 88.28 | 8888 | 249.21 | 22.91 |
| Jun | 7.90 | 5.32 | 86.29 | 13,154.4 | 181.04 | 22.82 |
| Jul | 7.39 | 5.62 | 85.84 | 9508.2 | 220.51 | 24.87 |
| Aug | 8.16 | 5.82 | 84.41 | 9272.3 | 345.55 | 26.76 |
| Sep | 9.79 | 4.27 | 80.64 | 6181.5 | 550.76 | 25.48 |
| Oct | 11.23 | 3.65 | 78.03 | 5741.9 | 801.51 | 28.82 |
| Nov | 12.99 | 3.95 | 76.44 | 4174 | 1019.96 | 28.82 |
| Dec | 14.93 | 3.01 | 72.87 | 3929.3 | 1237.38 | 25.16 |
| Average | 11.84 | 3.92 | 78.92 | 6210.0 | 677.04 | 23.67 |

Table 2 shows the statistical parameters from in situ measurements at six monitoring stations during the 2000–2021 period. At the six sampling stations, the behavior of SD varied from 6.46 m to 9.69 m during spring and summer, respectively. The minimum value of the SD was reported at stations V3, V5, and V6 (4.00 m) in spring, while the maximum

value, denoting high clarity in the water column, was found at station V5 (18.00 m) in summer. Similarly, the average KdPAR values ranged from 0.21 m$^{-1}$ (summer at station V5) to 0.31 m$^{-1}$ (spring, at stations V1 and V5). During this period maximum Chl-a values above 5.40 µg/L were found; in summer sampling stations V5 and V6 presented values of up to 20.03 µg/L, characteristic of eutrophic lakes [57]. In several instances, algal blooms were reported in Lake Villarrica [6]. This could have been related to the intense human activity in the surrounding watershed.

**Table 2.** Descriptive statistics of water quality parameters for Lake Villarrica during summer and spring.

| Parameter | Statistic | Summer | | | | | | Spring | | | | | |
|---|---|---|---|---|---|---|---|---|---|---|---|---|---|
| | | V1 | V2 | V3 | V4 | V5 | V6 | V1 | V2 | V3 | V4 | V5 | V6 |
| SD | min (m) | 4.5 | 4.5 | 4.5 | 4.5 | 5 | 4.5 | 4.5 | 4.5 | 4 | 5.5 | 4 | 4 |
| | max (m) | 16.5 | 11.5 | 14 | 15.5 | 18 | 11.5 | 12.5 | 11.5 | 13 | 11.5 | 12 | 11.5 |
| | Average | 7.61 | 7.62 | 8.56 | 9.69 | 9.17 | 7.5 | 6.5 | 7.68 | 7.89 | 8.31 | 6.46 | 7.5 |
| | SD | 2.48 | 2.04 | 3.16 | 3.65 | 3.58 | 2 | 2.54 | 2.01 | 2.47 | 1.58 | 2.53 | 2 |
| | CV (%) | 32.6 | 26.8 | 36.96 | 37.65 | 39.03 | 26.62 | 33.81 | 26.18 | 31.3 | 19.01 | 33.95 | 26.64 |
| | *n* | 53 | 74 | 26 | 13 | 23 | 36 | 29 | 73 | 31 | 16 | 38 | 37 |
| Chl-a | min (µg/L) | 0.21 | 0.2 | 0.45 | 0.24 | 0.84 | 1.67 | 0.1 | 0.24 | 0.24 | 0.52 | 0.14 | 0.3 |
| | max (µg/L) | 18.88 | 14.43 | 5.4 | 9.53 | 20.03 | 20.03 | 19.17 | 14.88 | 9.01 | 19.39 | 19.18 | 9.78 |
| | Average | 6.34 | 2.32 | 3.65 | 3.35 | 3.63 | 5.32 | 2.74 | 2.27 | 1.67 | 2.83 | 2.1 | 2.18 |
| | SD | 5.92 | 2.54 | 5.4 | 2.57 | 4.38 | 4.57 | 4.46 | 2.44 | 1.82 | 4.82 | 3.26 | 1.92 |
| | CV (%) | 93.43 | 109.57 | 148.04 | 76.62 | 120.59 | 85.85 | 162.67 | 107.44 | 109.33 | 170.17 | 154.93 | 88.03 |
| | *n* | 34 | 57 | 19 | 12 | 18 | 20 | 29 | 60 | 27 | 21 | 31 | 29 |
| Turbidity | min (NTU) | 0.1 | 0.1 | 0.34 | 0.1 | 0.1 | 0 | 0.24 | 0.2 | 0.1 | 0.1 | 0.1 | 0 |
| | max (NTU) | 2.42 | 4.7 | 6.17 | 0.92 | 4.8 | 2.5 | 7.86 | 5.01 | 3.39 | 0.4 | 3.81 | 3.5 |
| | Average | 0.94 | 1.11 | 1.29 | 0.52 | 0.77 | 0.86 | 2.3 | 2.22 | 1.02 | 0.22 | 1.32 | 1.38 |
| | SD | 0.67 | 1.23 | 1.61 | 0.21 | 1 | 0.67 | 3.05 | 1.69 | 1.25 | 0.15 | 1.3 | 1.17 |
| | CV (%) | 71.05 | 111.06 | 124.99 | 40.13 | 129.73 | 77.81 | 132.76 | 76.29 | 122.64 | 70.42 | 98.44 | 84.21 |
| | *n* | 39 | 37 | 22 | 15 | 22 | 26 | 19 | 29 | 23 | 10 | 23 | 23 |
| KdPAR | min (m$^{-1}$) | 0.12 | 0.17 | 0.14 | 0.13 | 0.11 | 0.17 | 0.16 | 0.17 | 0.15 | 0.17 | 0.17 | 0.17 |
| | max (m$^{-1}$) | 0.44 | 0.5 | 0.44 | 0.44 | 0.4 | 0.44 | 0.44 | 0.44 | 0.5 | 0.36 | 0.44 | 0.5 |
| | Average | 0.26 | 0.26 | 0.23 | 0.21 | 0.22 | 0.27 | 0.31 | 0.26 | 0.25 | 0.24 | 0.3 | 0.27 |
| | SD | 0.08 | 0.08 | 0.1 | 0.11 | 0.09 | 0.07 | 0.1 | 0.08 | 0.15 | 0.05 | 0.1 | 0.08 |
| | CV (%) | 30.52 | 29.72 | 43.15 | 51.81 | 40.23 | 26.15 | 31.4 | 30.09 | 61.54 | 22.77 | 32.6 | 30.53 |
| | *n* | 53 | 74 | 25 | 13 | 23 | 36 | 29 | 73 | 31 | 20 | 38 | 37 |

CV—coefficient of variation, *n*—data number, and SD—standard deviation.

Although this region does not have a dry season, there is a clear decrease in precipitation and wind in summer (see Table 1), such that the water column appears more stable and less mixed. This is demonstrated by the turbidity values, which reach minimum values of 0.52 NTU and 0.77 NTU in summer at stations V4 and V5, respectively. Increased turbidity values are reported in spring at stations V1 (2.30 NTU) and V2 (2.22 NTU), near the coast of the city of Villarrica, and station V6 (1.38), near Pucón. These high turbidity values may be related to runoff and contributions from the basin in the rainy months (see Table 1), as well as untreated sewage discharges and the growth of a highly populated area along the southern coast that does not have a sewage system and discharges waste directly without treatment [40].

## 3.2. Algorithms for KdPAR Estimation

Figure 2 shows the run of the four algorithms using L8/OLI and S2A/MSI images for the same day (14 March 2020). Values greater than 0.8 m$^{-1}$ (using QAA v6 KdPAR Lee) are presented near the coasts and population centers (Villarrica and Pucón), while the lowest are found in the central part of the lake and areas of less human activity. It can be observed that both satellites show a similar spatial distribution of KdPAR in the lake. This suggests that it is possible to use both satellites for the study of this water feature. It is important to

consider the combination of temporal frequency of both satellites, which would allow a greater number of images for the studied period to be obtained.

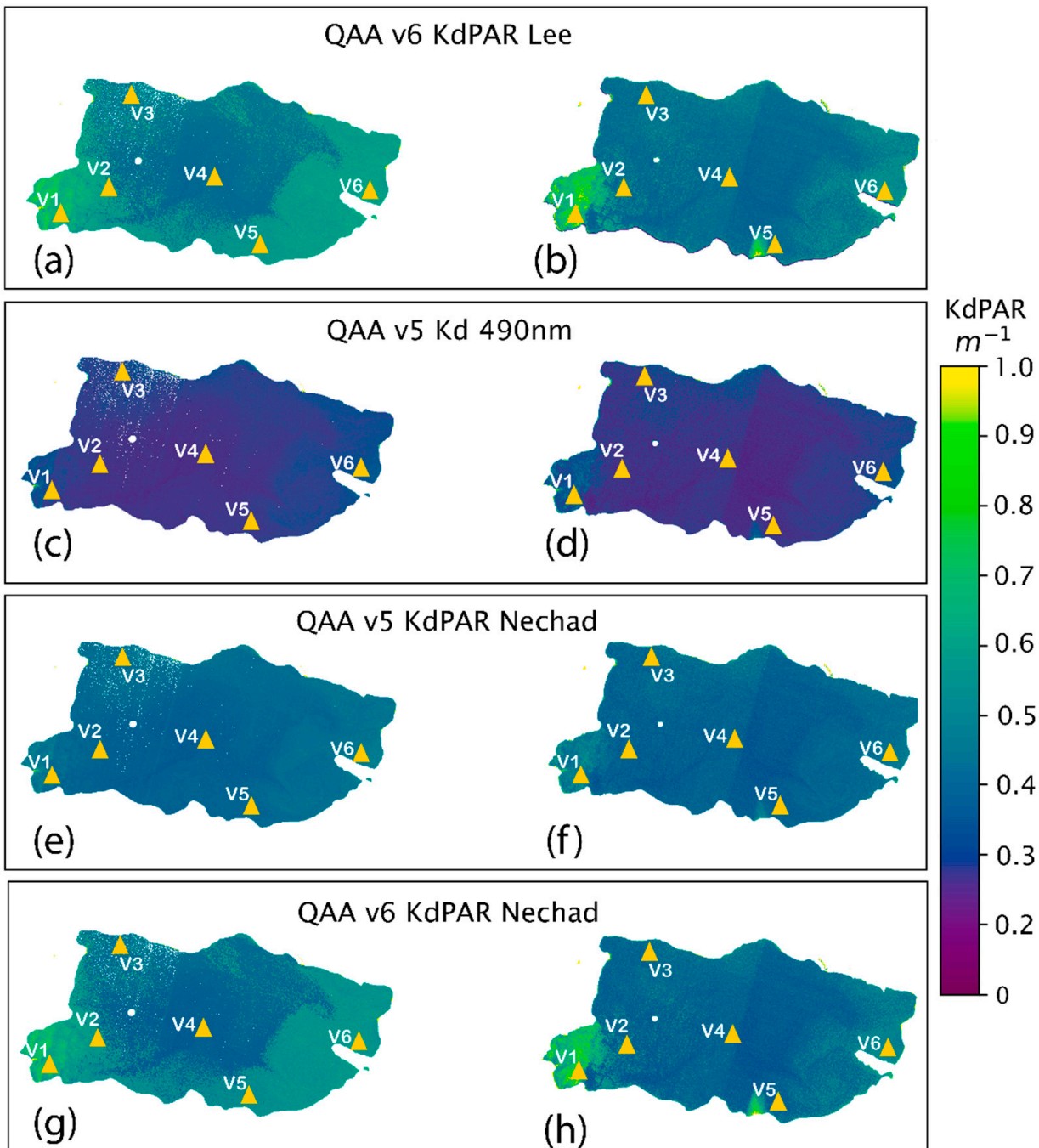

**Figure 2.** Estimation of KdPAR in Lake Villarrica using four algorithms in (**a**,**c**,**e**,**g**) L8/OLI and (**b**,**d**,**f**,**h**) S2A/MSI images from 14 March 2020.

### 3.3. Algorithm Validation

The analysis revealed that the coefficient of determination was large ($R^2 \geq 0.867$) when the selected algorithms were used in both seasons, which are shown in Figure 3. Indicating a relatively good level of agreement between measured and estimated KdPAR values. The v6 algorithm tends to fit a positive exponential function, departing from the 1:1 line, while v5 algorithm is a relatively closer fit to the 1:1 line. The *MAPE* ranged

between 25.9–93.9% (QAA v5 KdPAR Nechad-QAA v6 KdPAR Lee) for spring and summer, respectively. Based on the *MAPE* indicator, the worst results with significantly different values from their measured counterparts were presented by QAA v6 KdPAR Lee. Similarly, the highest *RMSE* (0.19 m$^{-1}$) was found in summer using the QAA v6 KdPAR Lee and QAA v6 KdPAR Nechad algorithms, while the lowest *RMSE* (0.02 m$^{-1}$) was found in spring using the QAA v5 KdPAR Nechad algorithm. The average bias (positive result in most cases) ranged between 0.023 m$^{-1}$–0.234 m$^{-1}$, with higher over-estimates (0.157 m$^{-1}$ and 0.234 m$^{-1}$) found using the QAA v6 KdPAR Lee in spring and summer, respectively. This is probably because in the case of Lee's algorithm version 6 a wavelength reference of 667 nm was proposed for water with Rrs(670) > 0.0065 sr$^{-1}$, i.e., water with higher concentrations of Chl-a [30]. Thus, the algorithm version 6 is not suitable for Lake Villarrica. Therefore, results indicate that the algorithm version 5 can provide reliable/fair predictions of KdPAR during summer and spring.

To select the best algorithm during summer and spring, a Taylor diagram was drawn. Figure 4 illustrates the seasonal evaluation of the algorithms and shows a good correlation; lowers values of standard deviation and *RMSE* between reference data and estimated KdPAR with QAA v5 Kd490 nm for summer. A similar result was found in spring, where QAA v5 KdPAR Nechad was the best algorithm for KdPAR estimation. The worst-performing algorithm was obtained in spring using QAA v6 KdPAR Nechad, with values of 0.54 and 0.22 m$^{-1}$ (lower correlations and higher *RMSE*).

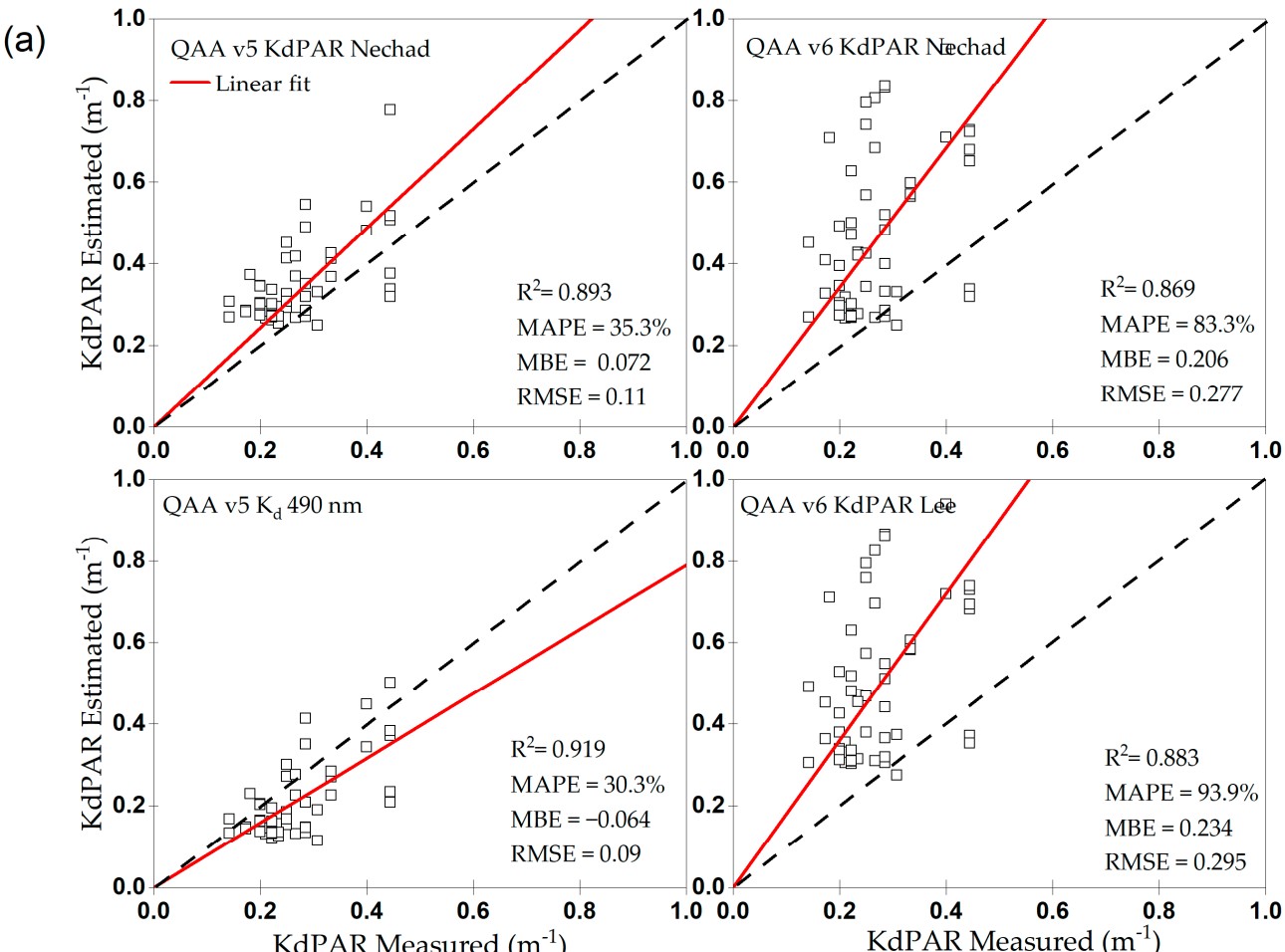

**Figure 3.** *Cont.*

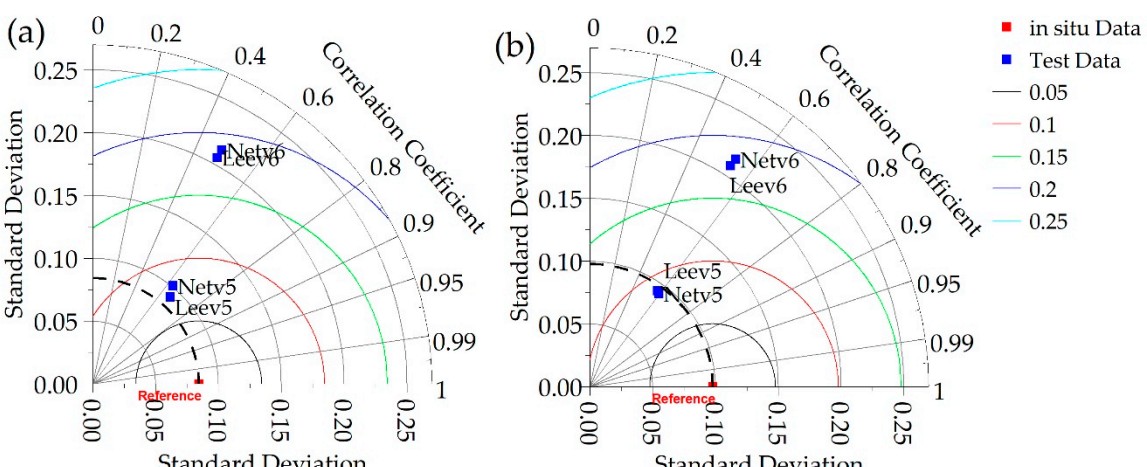

**Figure 3.** Estimated and measured KdPAR using different algorithms during (**a**) summer (*n* = 55) and (**b**) spring (*n* = 43), *n* represents the sample size. The dashed black line is the 1:1 line. Red lines indicate the fitting using the least-squares regression method.

**Figure 4.** Comparison of the performance of the algorithms used for the two study seasons—(**a**) summer and (**b**) spring—using a Taylor diagram. Where: Netv5 (QAA v5 KdPAR Nechad), Netv6 (QAA v6 KdPAR Nechad), Leev5 (QAA v5 Kd 490 nm), and Leev6 (QAA v6 KdPAR Lee).

### 3.4. Seasonal Spatial Pattern of KdPAR

Figure 5 shows the map and intralake variations of the KdPAR parameter in Lake Villarrica using the best algorithm.

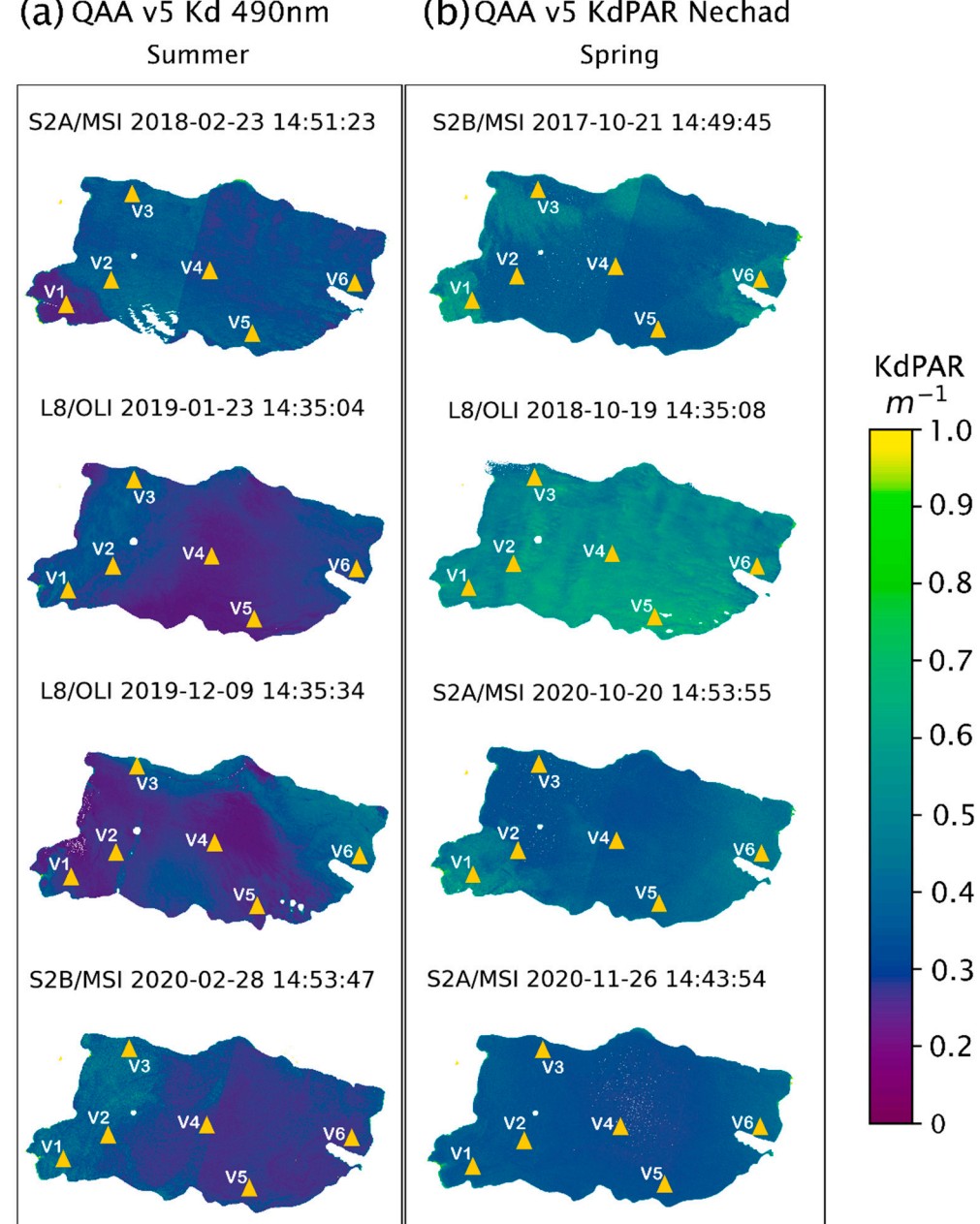

**Figure 5.** Seasonal spatial pattern of KdPAR estimation using (**a**) QAA v5 Kd490 nm and (**b**) QAA v5 KdPAR Nechad during summer and spring in Lake Villarrica.

In general, sites near the shoreline (Villarrica and Pucón) presented the maximum values for each processed image. The highest KdPAR values were distributed near the river mouths (e.g., Pucón, Tolten, and Trancura rivers). This indicates that heavy rainfall in the river basin will carry amounts of sediment to Lake Villarrica. Some events are consistent with reports of the DGA [58]; however, most of these river stations are not official.

The maximum values are mainly estimated to be between 0.526–0.608 m$^{-1}$ (V4,V5) in spring (19 October 2018) and 0.517–0.545 m$^{-1}$ (V3–V1) in summer (28 February 2020). The central sampling station (V4) always presents the lowest levels of KdPAR in comparison to other regions of the lake, with exceptions on 23 February 2018 and 19 October 2018. These

values are likely due to local conditions such as episodes of Puelche wind and precipitation days before imaging and sampling. The relationship between KdPAR and meteorological conditions is explained in detail in Section 3.6.

### 3.5. KdPAR and Its Relationship with Meteorological Variables

To further investigate the contribution of several meteorological parameters to the estimation of KdPAR, a qualitative analysis was performed. Figure 6 shows the selected satellite days, representing spring and summer. On the selected days the predominant wind direction was toward the SW and NW, reaching a maximum speed of 3.4 ms$^{-1}$ (28 February 2020), while the lowest daily mean corresponded to a light westerly breeze of 0.88 ms$^{-1}$, mostly toward the SW (26 November 2020). The selection of days 28 February 2020 and 19 October 2018 provided evidence that episodes of strong wind, perhaps Puelche wind, can force significant mixing in the water column, resulting in a change in the thermal structure of the lake and the vertical transport of nutrients from the thermocline, which can increase lake productivity [40] and KdPAR values. Therefore, some high KdPAR values are associated with strong wind due to sediment resuspension. Thus, Lake Villarrica may be more vulnerable to eutrophication resulting from these local meteorological conditions. In the satellite image from 19 October 2018, KdPAR distribution was observed after days of heavy rainfall, which totaled 50 mm. This rainfall, together with wind speeds higher than 2 ms$^{-1}$ and gusts of 6 ms$^{-1}$, can influence the distribution of KdPAR, making it practically uniform throughout the lake.

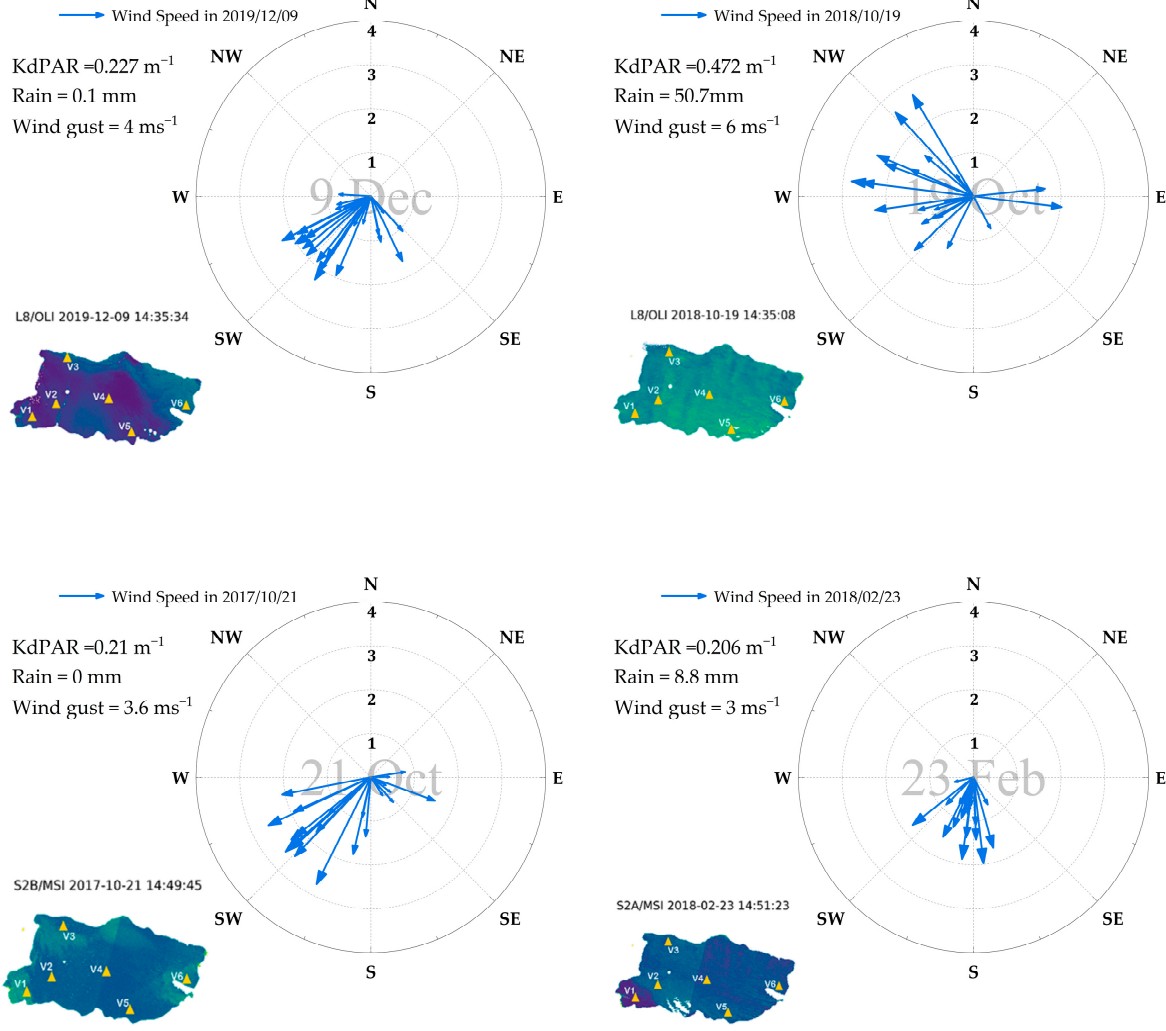

**Figure 6.** *Cont.*

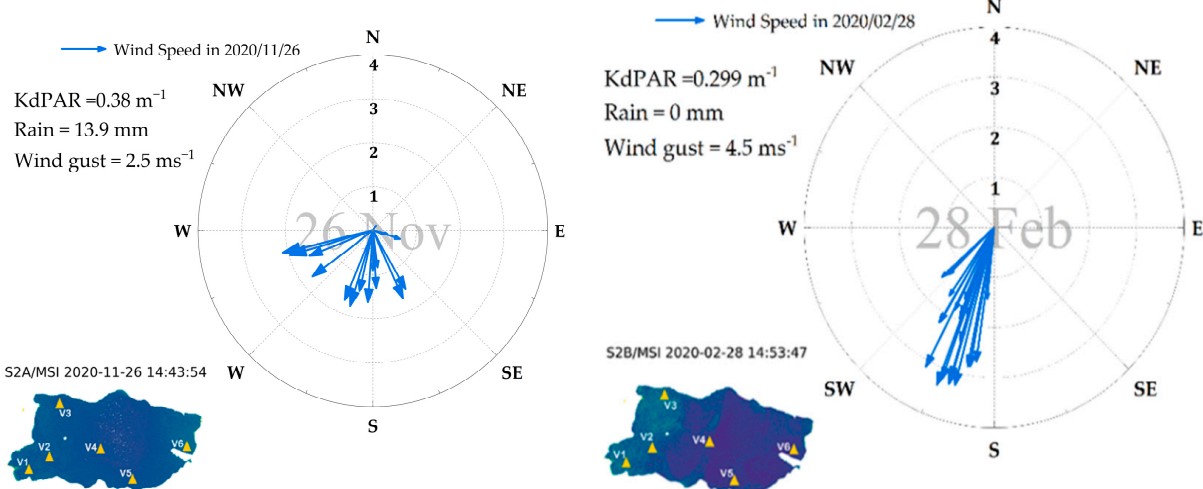

**Figure 6.** KdPAR and its relationship with wind speed, gust speed, wind direction, and rain on selected days.

### 3.6. Optical Water Classification

Transparency values (meters) were recorded, measured by SD in several monitoring campaigns to ascertain the behavior of the passage of light through the water column, and later used to calculate the KdPAR. The values obtained in the time series (2000–2021) for each sampling station were averaged by season (see Table 2). The highest values are observed in spring due to possible mixing in the water column, which could cause turbidity that would prevent the passage of light. The greatest transparency is found in summer because of the characteristics of the station itself. In summer it rains less, the waters are calmer, so the water column is more transparent, and the light reaches a greater depth. In addition, in this period the lake is stratified, so there is little water circulation, and the equilibrium allows greater passage of light through the water.

According to the obtained PAR attenuation coefficients, Lake Villarrica behaves similarly to Jerlov's oceanic water type III and coastal water C1 (see Table 3). In all cases, the optical classification of the lake is consistent with the trophic characteristics of a mesotrophic aquatic system (clear water with intermediate productivity levels) [59]. It can be suggested that it has remained in a meso-oligotrophic state. This is probably linked to the occurrence of some algal bloom events [6], which could be more frequent in the summer season, when the lake is under greater pressure due to the increase of the floating population. These blooms have a direct impact on the optical quality of the water and often cause serious economic and ecological impacts, especially in coastal areas, while threatening human and marine health. This result is useful in bio-optical studies, to address the variability in biological processes (e.g., primary production in Villarrica Lake), and identify optical behaviors in the lake that deviate from the mean trend, and water quality dynamics. Furthermore, the results show the synergy between SD (or inferred KdPAR) and water type, so Jerlov's work was extended to freshwater ecosystems. Thus, an optical classification was proposed for the Villarrica freshwater ecosystem that can be extended to other lake bodies.

**Table 3.** Optical classification of Jerlov's oceanic and coastal water and SD-derived KdPAR for Lake Villarrica. Source: [19].

| Optical Classification | Coastal Water Type | | | | | Ocean Water Type | | | Lake Villarrica | |
|---|---|---|---|---|---|---|---|---|---|---|
| | C1 | C3 | C5 | C7 | C9 | I | II | III | C1, III | C1, III |
| KdPAR (m$^{-1}$) | 0.29 | 0.38 | 0.51 | 0.71 | 1.04 | 0.15 | 0.19 | 0.25 | 0.24 Summer | 0.27 Spring |

## 4. Discussion

Water transparency is an important property of a freshwater lake ecosystem. Increases or decreases in water transparency can negatively affect the biological component of the system, which may be adapted to specific light penetration conditions [18]. Our findings indicate that the physical and biological characteristics of the aquatic system of Lake Villarrica vary seasonally and that these changes influence KdPAR. The KdPAR of the aquatic system Lake Villarrica can be derived from in situ measurements of SD depth and satellite data through different algorithms, which is an advantage given the affordability of the method. The spatial values of KdPAR obtained from satellite images reflect a difference between seasons. A greater number of satellite images are available for summer than for spring. In the latter, cloud cover becomes a determining factor for the selection of images. Therefore, it is suggested that remote sensing studies be complemented with manned devices (drones) in seasons in which cloudiness is a determining factor—such as the austral winter—or with radar images. This last approach will be analyzed in future research. For this study, the multispectral images obtained from Landsat and Sentinel satellites complemented each other, showing that the PAR attenuation coefficient estimation algorithms presented similar results for Lake Villarrica.

The KdPAR estimated from QQA algorithms was validated with KdPAR derived from the SD in situ measurements at six sampling stations, presenting reliable/fair predictions. The best results were obtained with QAA v5KdPAR Nechad ($R^2$ = 0.931, $MBE$ = 0.023 m$^{-1}$, $RMSE$ = 0.088 m$^{-1}$, and $MAPE$ = 35.9%) for spring and QAA v5 Kd490 algorithms ($R^2$ = 0.919, $MBE$ = −0.064 m$^{-1}$, $RMSE$ = −0.09 m$^{-1}$, and $MAPE$ = 30.3%) for summer. This was mainly because the empirical steps involved in this algorithm were suitable for type III waters with less complex optical properties, resulting in smaller errors in the estimation of the inherent optical properties that were eventually passed on to the estimation of KdPAR. As a result, the estimation accuracy of QAA v5 was higher than that of QAA v6. Therefore, application of the algorithms with the best results for Lake Villarrica is a useful tool not only to ascertain the spatial variation of this variable, but also to show intralake differences.

Based on Jerlov delineations, the water of Lake Villarrica can be optically classified as oceanic type III and coastal type C1, which is consistent with the trophic state classification attributed to Lake Villarrica [60]. According to the physical, chemical, and biological characteristics of the lake it is mesotrophic, a state of transition from oligotrophic to eutrophic, a natural process that all aquatic systems undergo, but which has been accelerated in Lake Villarrica by the intensive development and tourist activity in its basin [43]. The cities of Villarrica and Pucón are important tourism centers in southern Chile, the populations of which have increased in the last decade [61]. If the effects of urbanization are added to the effects of climate change, such as changing hydrodynamic conditions, precipitation, and other factors in most of Chile's lakes, those located next to cities will be more affected by these multiple factors. For example, Lake Riñihue has average KdPAR values of 0.16 m$^{-1}$ and 0.15 m$^{-1}$ for spring and summer, much lower than those reported in this work for Villarrica [62].

The results indicate that precipitation and wind are the most influential factors in water transparency and the PAR radiation attenuation coefficient. Spring presented higher precipitation values, which affect the physical parameters in the water column; in periods of rain and strong winds, turbidity in the water column increases, generating mixing and thus limiting the passage of light and increasing the attenuation coefficient in the water column at depth. During summer, a period when hydrometeorological variables have less influence, allochthonous inputs from the basin predominate. Thus, the highest turbidity values and lowest transparency and KdPAR values are reported in the monitoring stations closest to the population centers of Lake Villarrica, mainly the cities of Villarrica and Pucón. The chain of lakes to which Lake Villarrica belongs, the Araucanian lakes, is a series of interconnected lakes with high cultural, recreational, and scenic value, and the water quality of these lake systems reflects their natural value

and their high economic, environmental, and social importance. Lake Villarrica is the fourth largest lake in the Araucanian chain and has a water circulation renewal time of 4 years [42]. Thus, disturbances in this body of water affect its transparency and affect the ecosystem services associated with its use. The Chilean government maintains a secondary water quality standard for Lake Villarrica [41], which has been exceeded due to several phenomena reported in the lake such as algal blooms [6]. The next step will be to examine the links between the results obtained in this study on the temporal and spatial behavior of water transparency and events that affect transparency such as algal blooms, which are increasingly frequent in this southern lake.

## 5. Conclusions

This work used 36 satellite images (L8/OLI and S2A/MSI) to evaluate four algorithms that retrieve KdPAR at six sampling stations in Lake Villarrica, Chile.

The best results were obtained with QAA v5 KdPAR Nechad for spring and QAA v5 Kd490 algorithms for summer.

Spatio-temporal distribution of KdPAR values was related to local meteorological conditions.

Based on SD-derived KdPAR, the lake was related to a specific optical water type using Jerlov's classification.

**Supplementary Materials:** The following supporting information can be downloaded at: https://www.mdpi.com/article/10.3390/rs14184568/s1, Table S1: Operational characteristics of radiometric sensors; Table S2: Landsat and Sentinel image characteristics and in situ measurements; Figure S1: a–d: Estimation of KdPAR in Lake Villarrica using the four algorithms running in L8/OLI and S2A/MSI images.

**Author Contributions:** Conceptualization: L.R.-L. and L.G.-R.; methodology: L.R.-L., L.G.-R., I.D.-L. and W.G.; software: I.D.-L., W.G. and L.G.-R.; validation: L.R.-L. and L.G.-R.; formal analysis: L.R.-L.; investigation: R.U.; resources: R.U.; data curation: L.R.-L. and L.G.-R.; writing—original draft preparation: L.R.-L. and L.G.-R.; writing—review and editing: L.R.-L., I.D.-L. and L.G.-R.; visualization: L.G.-R.; supervision: R.C.; project administration: L.R.-L. and R.U.; funding acquisition: L.G.-R., R.U. and L.R.-L. All authors have read and agreed to the published version of the manuscript.

**Funding:** This research was funded by Universidad de las Américas and the CRHIAM Water Center (ANID/FONDAP/15130015).

**Data Availability Statement:** The data presented in this study are available on request from the corresponding author.

**Acknowledgments:** L.R.-L. is grateful to the VRIDFAI21/10 project of the Universidad San Sebastian. Special thanks to the Centro de Recursos Hídricos para la Agricultura y la Minería (CRHIAM) (Project ANID/FONDAP/15130015). L.G.-R. thanks the Universidad de las Américas, Chile (UDLA). I.D.-L. thanks the Hémera Centro de Observación de la Tierra of the Universidad Mayor. All authors are grateful for the online monitoring networks and satellites, such as those of the Chilean Meteorological Directorate (DMC), the National Institute of Agricultural Research (INIA), and Landsat and Sentinel programs.

**Conflicts of Interest:** The authors declare no conflict of interest.

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
