# Peer review of "Assessment of the Diffuse Attenuation Coefficient of Photosynthetically Active Radiation in a Chilean Lake"

_remotesensing, doi:10.3390/rs14184568_

Round 1
Reviewer 1 Report
The presented work is of undoubted importance and value for the study of light fields in the lakes of Chile. Of particular value is the extensive historical array of traditional in-situ data and the resulting comparison of in-situ and high-resolution satellite optical data.
The work deserves to be published in the Remote Sensing journal after correcting the following comments:
1) It is necessary to add a mention of Lake Villarrica in the abstract and keywords.
2) It is necessary to clarify the method by which the concentration of chlorophyll-a were determined and the device for measuring turbidity.
3) Specify how OD was defined in formula (1) so that the final expression or algorithm corresponds to only in-situ measurements.
4) On line 251 and in Table 1, specify which temperature is meant – air or water?
5) In line 329, you write about good correlation values of 0.664-0.613, however, in all the graphs presented in Fig. 3, the coefficient of determination R2 is greater than 0.8. Specify or describe in more detail to understand about this discrepancy.
6) What value does the algorithm "QAA v5 Kd 490 nm" calculate? Kd490 or KdPAR? If the Kd490, then strictly speaking Kd490 and KdPAR are not the same thing and additional parameterizations are needed to convert one value into another, and your final conclusions may change.
Author Response
Responses to Reviewers
********
We thank the editor and reviewers for their extended time and the valuable comments, which improve the quality of our manuscript. Our responses are formatted in bold and cursive with the mark “//”. In the manuscript the changes and additions are highlighted in yellow, gray, blue and green colors, following the patterns Reviewer #1: yellow color, Reviewer #2: gray color, Reviewer #3 blue color and Reviewer #4: green color, respectively
Other changes (highlighted in bold)

Reviewer 2 Report
Overall Comment:
The manuscript is well written and smooth to follow with clear objectives and hypothesis. However, method section needs more clarity and supplementary table should be cited to know which satellite dates were used to match-up with in-situ data (if possible that table can be moved to the main text section). The satellite data match-up with in-situ data for up to +-8 days difference seem too long if the wind effect is dominant in the lake. How KdPAR was estimated from in-situ SD without OD was also not clear. Also, why authors derived only kdPAR from satellite when field data for Chl-a and turbidity was also available. Both Chl-a and turbidity can be derived using ACOLITE that authors have used in this study. Please justify. Overall, this is a good regional study but how this study contributes to international scientific community should be discussed as well.
Specific comments:
Fig. 1. Please mention which satellite image it is in the figure caption.
Line 177: “Thus KdPAR was calculated…..using Eq. 1” How did you get OD value? Because Eq. 1 has both OD and SD to estimate KdPAR.
Line 196: “In addition, 22 Sentinel 2 (S2 A/B) images were selected for this study.” What was the criteria to select these 22 Sentinel 2 images?
Statistical Analysis:
Table 4. I am surprised to see that sampling location V4 has highest average SD values among all sampling locations in Summer but has the highest average turbidity value as well. This was just opposite in case of Spring season. Can you please explain why?
Lines 295-296: “. It can 295 be observed that both satellites show a similar spatial distribution of KdPAR in the lake.” I can see clear underestimation in Sentinel 2 maps compared to Landsat maps, especially near the boundary of the lake. Can you please explain why? Also, there is a salt and pepper type of noise present in Landsat 8 derived maps, why? Problem with atmospheric correction? Please explain.
Fig. 2. There is a typo in caption. Should be “(a, c, e, g)” instead of ‘(a, c, d, g)”.
Supplementary section: The supplementary Table has typo “Dic” for “Dec”
Author Response

(The authors gave the same response as above.)

Reviewer 3 Report
This is a well written paper that is based on a substantial body of work by the authors. The authors analyzed in detail the history of KdPar over 19 years, and correlations of multispectral satellite derived data with in-situ data to establish the optimum algorithm set to retrieve the relevant parameters. The comprehensive statistical analysis the authors use allows them to make a convincing case. The results are extensively and precisely described and their interrelation and the environmental dynamics involved clearly studied and presented. Accurate and timely information with large area coverage about the biologically relevant optical parameters is an important ecological problem particularly for inland waters bodies with low water turnover. This contribution to it should definitely be published in my opinion.
However I have noted some issues which I believe should be addressed before publication to improve the paper. I have noted those problems along with minor deficiencies/improvements in my comments below.
Line 32: “event suggested by sediment resuspension” should be replaced by:”events suggesting they are caused by sediments resuspension.”
Line 182 Equation 1 define precisely OD and SD. Optical depth at what wavelength or wavelength band and is it to base 10 or to base e.
Line 336 It is not clear in the Taylor diagram what the kdpar point refers to. An explanation of why it is a reference if such is the case should be added to the figure caption.
Line 413 “Although the KdPAR models used presented good KdPAR estimation results for Villarrica, it is better to use algorithms for specific water types such as Chl-a, turbidity, etc. because it varies greatly in inland waters”. The statement is unclear, should one use algos for chlorophyll and turbidity and then estimate KdPAR from those results? If so you should describe the relationships one needs to use to proceed from chlorophyll and turbidity to KdPAR.
Author Response

(The authors gave the same response as above.)

Reviewer 4 Report
A review of “Assessment of the diffuse attenuation coefficient of photosynthetically active radiation in a Chilean lake” authored by Lien Rodríguez-López, Lisdelys González-Rodríguez, Iongel Duran-Llacer, Wirmer García, Rolando Cardenas, and Roberto Urrutia
Summary: the authors used 14 Landsat-8 and 22 Sentinel-2 images over Lake Villarrica in order to evaluate 4 algorithms predicting Kd (QAA v5 Kd490 nm; QAA v6 KdPAR Lee; QAA v5 KdPAR Nechad; QAA v6 KdPAR Nechad). The best results were obtained with QAA v5 KdPAR Nechad for spring and QAA v5 Kd490 algorithm for summer. Variability in Kd over the lake was related to meteorological variability. Based on Secchi disc depth values, the lake was also classified using Jerlov delineations.
Major comments:
The authors did not measure Kd, they measured the Secchi disc depth. Why did they convert these data to Kd? They could have evaluated Lee’s Secchi disc depth model (https://www.sciencedirect.com/science/article/pii/S0034425715300900). Secchi disc depth is a very useful indicator of ecosystem health/lake trophic status, why not use these data directly?
Kd is an ‘apparent’ optical property (AOP) as it is angle-dependent.
There are issues throughout the document with defining a term (e.x., Kd), then keeping annotation consistent throughout. I would define Kd(PAR) or Kd(lambda) initially, then keep that notation consistent. Once a term is defined, the abbreviation can be used. Another example is units. The way the units are presented needs to be consistent throughout the document.
I think that there needs to be more discussion on the optical classification results. Why is this useful? How can this information be used practically in Lake Villarrica?
Lns 225-226/Figure 2: why was not only the best algorithms used to evaluate spatio-temporal patterns in Kd? I thought the side-by-side images of Landsat/Sentinel were very interesting, though.
Ln 221-222: I think more explanation about the models, might be useful. Are they (for Kd) IOP-based, empirical? If empirical, what is the range of data over which the authors used to develop their models? This could be also be included in the Discussion, I.e., why some models performed better than others.
Ln 296-297: This is a qualitative statement. Can the authors turn this in to a quantitative analysis? How do the two satellite-based values for Kd compare for each algorithm for each site?
Figure 3. Is the sample size (n=) listed anywhere? Maybe I missed this.
Lns 327-336: I think the authors need to describe how the Taylor diagrams are used in conjunction with the previous regression analysis. How do the results from the Taylor diagrams and regression analyses compare to each other?
Ln 358: the authors say a “simple analysis was performed”. What analysis? The authors needs to describe this in more detail. It seems like a qualitative analysis is decried below.
Ln 385-387: the wording is awkward here.
“Conclusions’ is more like a Summary.
Minor comments:
Lns 48-49: solar radiation is crucial to nearly all of earth’s ecosystems.
Ln 79: long-term? Over what temporal scale?
Ln 86: ‘mean’?
Ln 129: I would remove “and it has an elliptical shape”.
Ln 304: Figure 3 isn’t revealing the results; the analysis revealed the results, which are shown in Figure 3.
Ln 411: in both what?
Ln 438-439: when turbidity increases, Kd increases.
Author Response

(The authors gave the same response as above.)

Round 2
Reviewer 1 Report
Dear authors!
Thanks for the edits and clarifications. The article can be published in the journal.